# The Potent Antitumor Activity of Smp43 against Non-Small-Cell Lung Cancer A549 Cells via Inducing Membranolysis and Mitochondrial Dysfunction

**DOI:** 10.3390/toxins15050347

**Published:** 2023-05-19

**Authors:** Ze Deng, Yahua Gao, Tienthanh Nguyen, Jinwei Chai, Jiena Wu, Jiali Li, Mohamed A. Abdel-Rahman, Xueqing Xu, Xin Chen

**Affiliations:** 1Department of Pulmonary and Critical Care Medicine, Zhujiang Hospital, Southern Medical University, Guangzhou 510280, China; stzbdz1991@163.com; 2Guangdong Provincial Key Laboratory of New Drug Screening, School of Pharmaceutical Sciences, Southern Medical University, Guangzhou 510515, China; gaoyahua7721@163.com (Y.G.); thanhnguyentien18992@gmail.com (T.N.); vivian20@i.smu.edu.cn (J.C.); jienawu1996@163.com (J.W.); lijiali6690@163.com (J.L.); 3Zoology Department, Faculty of Science, Suez Canal University, Ismailia 41522, Egypt; mohamed_hassanain@science.suez.edu.eg

**Keywords:** antimicrobial peptide, anticancer, Smp43, membranolysis, mitochondrial dysfunction

## Abstract

Research has been conducted to investigate the potential application of scorpion venom-derived peptides in cancer therapy. Smp43, a cationic antimicrobial peptide from *Scorpio maurus palmatus* venom, has been found to exhibit suppressive activity against the proliferation of multiple cancer cell lines. However, its impact on non-small-cell lung cancer (NSCLC) cell lines has not been previously investigated. This study aimed to determine the cytotoxicity of Smp43 towards various NSCLC cell lines, particularly A549 cells with an IC_50_ value of 2.58 μM. The results indicated that Smp43 was internalized into A549 cells through membranolysis and endocytosis, which caused cytoskeleton disorganization, a loss of mitochondrial membrane potential, an accumulation of reactive oxygen species (ROS), and abnormal apoptosis, cell cycle distribution, and autophagy due to mitochondrial dysfunction. Additionally, the study explored the in vivo protective effect of Smp43 in xenograft mice. The findings suggest that Smp43 has potential anticarcinoma properties exerted via the inducement of cellular processes related to cell membrane disruption and mitochondrial dysfunction.

## 1. Introduction

A great deal of attention has been paid to lung cancer treatment due to this being the leading cause of cancer-related death worldwide. Among the histologically classified types, non-small-cell lung cancer (NSCLC) accounts for approximately 75–80% of all lung cancer cases [1]. The treatment options for NSCLC might include surgery, radiofrequency ablation, radiation therapy, and chemotherapy. For patients who are not suitable for surgery, combined chemotherapy and radiation therapy may become the primary treatment. However, due to the non-selective cytotoxicity and drug resistance associated with conventional therapies, there is an urgent need for alternatives with fewer side effects in lung cancer treatment.

As a part of the innate immune response, antimicrobial peptides (AMPs) have been found to possess multiple functional activities apart from killing microorganisms, such as antiviral activities, anticancer activities, immunomodulation, and so on [2,3,4]. AMPs are characterized by net positive charges and hydrophobic residues, accounting for 30% to 60% of total residues, making them suitable for selective cytotoxicity against cancer cells due to the excessive anionic constituents of cell membranes compared to those of normal cells [5]. The strong binding reaction between AMPs and cancer cells, caused by electrostatic attraction, is crucial for the membranolytic mode of action of AMPs [6]. Furthermore, AMPs exert their antitumor effect by targeting other cellular structures or selectively interacting with ion channels [7,8]. The unique cytotoxic mechanism of AMPs against cancer cells shows potential for solving the multi-drug resistance problem caused by conventional therapies.

Due to their environmental adaptation, self-protection, or predatory behavior, venomous animals produce peptides, either by themselves or through their venom gland, that have become a rich source for drug development, especially in anticancer therapy [9]. Scorpion venom-derived peptides have been shown to inhibit the proliferation of cancer cells by blocking voltage-dependent ion channels, disrupting cell membranes, or interfering with the cellular process by targeting cellular structure, making them promising antitumor candidates. For instance, MgTX, derived from the toxin of *Centruroides margartarus*, suppresses the viability of A549 cells both in vivo and in vitro by blocking voltage-gated K^+^ channels [10]. Previous studies have also demonstrated the anticancer activity of *Scorpio maurus palmatus*-derived AMP Smp24 exerted via the inducement of membranolysis and mitochondrial dysfunction [11,12]. In this study, we investigate the effect of the inhibitory capability of Smp43 (GVWDWIKKTAGKIWNSEPVKALKSQALNAAKNFVAEKIGATPS) on the proliferation of NSCLC A549 cells, another *S. m. palmatus* venom-derived peptide that has been explored for its antitumor activity against acute leukemia cell lines and human hepatoma HepG2 cells [13,14]. Our findings reveal that the effect of the antitumor capability of Smp43 on A549 cells is associated with membrane disruption, the disorganization of the cytoskeleton, ROS accumulation, apoptosis, abnormal cell cycle distribution, and autophagy induced by mitochondrial dysfunction.

## 2. Results

### 2.1. Smp43 Reduces Cell Viability in Human Lung Cancer

The suppressive capability of Smp43 towards different human lung cancer cell lines and human fetal lung fibroblast cells, MRC-5 cells, was evaluated using a MTT assay. As presented in Figure 1A, the viability of A549, PC9, H3122, H460, and MRC-5 cells was concentration-dependently suppressed in the presence of Smp43 with IC_50_ values of approximately 2.58, 7.74, 3.28, 7.23, and 12.26 μM, respectively. According to these data, A549 cells were chosen for further experiments to examine the antitumor capability of Smp43. Indeed, Smp43 significantly suppressed the proliferation of A549 cells in both a dose- and time-dependent manner (Figure 1B). The inhibitory effect of Smp43 against A549 cells was further confirmed via an EdU staining assay. Consistently, the quantity of A549 cells indicated by green fluorescence gradually decreased with a rising concentration of Smp43 (Figure 1C). The changes in the morphology of A549 cells treated with different concentrations of Smp43 can be observed in Figure 1D. In comparison to the untreated cells possessing normal fusiform shapes with a clear boundary and smooth surface, Smp43 treatment led to significant changes, including a decrease in the volume of A549 cells with a rounder shape. Moreover, floated cells and cellular debris were also observed. In agreement, compared to the untreated control cells, Smp43 concentration-dependently inhibited the colony formation of A549 cells (Figure 1E).

### 2.2. Smp43 Is Internalized into the A549 Cells via Endocytosis

The variation in the cell surface charge caused by Smp43 was calculated because the cationic peptide usually exerts its inhibitory effect by interacting with the cell membrane. As shown in Figure 2B, Smp43 markedly increased the A549 cell’s zeta potential from −9.62 mV to −3.24 mV in a concentration-dependent manner, indicating the interaction of Smp43 with the cancer cell membrane. Subsequently, the internalization of Smp43 into A549 cells was explored by evaluating the changes in the fluorescence of FITC-labeled Smp43. Indeed, the visible increase in the fluorescence of the labeled peptide-treated cells indicated the internalization of Smp43 into A549 cells (Figure 2A). The accumulation of Smp43 was significantly observed in the cytoplasm of these cells after 6 h of treatment, when compared to the control cells, and subsequently concentrated in the cell nucleus at 12 and 24 h. Due to the contribution to the interaction between a cell-penetrating peptide and the cell membrane [15], A549 cells were incubated with heparin to explore whether or not this affected the cellular uptake of Smp43. As displayed in Figure 2C, pre-treatment of heparin reduced the internalization of Smp43 into A549 cells. Moreover, the cellular uptake of Smp43 by the A549 cells at 37 °C was increased relative to that at 4 °C, indicating a thermo-sensitive and energy-dependent manner in the internalization of Smp43 into the A549 cells. Notably, the cellular uptake of Smp43 was markedly suppressed in the presence of the endocytosis inhibitor ammonium chloride, compared with that of the control group. These findings suggest a correlation between endocytosis and the internalization of Smp43.

### 2.3. Smp43 Induces Cell Membrane Disruption and Changes Cell Cytoskeleton Organization

As presented in Figure 3A, the LDH levels released from A549 cells were significantly increased in both a time- and dose-dependent manner in the presence of Smp43, indicating cell membrane leakage. Thus, the changes in the cell membranes caused by Smp43 were observed via a SEM. Compared to the well-defined shape and the smooth surface of the cell membranes of control cells, visible changes such as rough boundaries of the cell membrane and a leakage of cellular contents were observed in the Smp43-treated cells (Figure 3B). Due to the crucial role in multiple cellular processes such as regulating cell shape, motility, attachment, division, and cell signaling [16], the changes in the actin filament (F-actin) caused by Smp43 were investigated via rhodamine–phalloidin staining. As shown in Figure 3C, the treatment with Smp43 led to the visible disassembly of the F-actin structure, whereas the untreated cells displayed a well-patterned shape.

### 2.4. Smp43 Induces Mitochondrial Damage and ROS Overproduction in A549 Cells

Upon binding to cancer cells, AMPs might either disrupt the cell membrane or penetrate and damage mitochondria, leading to ROS accumulation, mitochondrial dysfunction, and eventual cell death. As presented in Figure 4A, Smp43 dose-dependently decreased the mitochondrial membrane potential of A549 cells, which was displayed by the increase in green fluorescence, accompanied by the decreased red fluorescence in the JC-1 staining assay. Moreover, pre-treatment with CsA may ameliorate the alteration of membrane potential caused by Smp43. The loss of mitochondrial membrane led to the overproduction of ROS, which was observed via the gradual increase in the green fluorescence intensity in the DCFH-DA staining assay (Figure 4B). It is noteworthy that pre-treatment with antioxidant NAC ameliorated the excessive secretion of ROS caused by Smp43 (Figure 4C). These findings suggest that Smp43 might damage the mitochondria, causing an alteration in mitochondrial membrane potential and the accumulation of ROS.

### 2.5. Smp43 Induces Apoptosis in A549 Cells

The A549 cell apoptosis caused by Smp43 was evaluated due to the fact that damaged mitochondria might affect the apoptosis process of cells by releasing multiple apoptotic factors [17]. As shown in Figure 5A, treatment with Smp43 caused a visible shrinkage of A549 cells as well as the formation of apoptotic bodies. Consistently, flow cytometry analysis revealed that the apoptotic cell rate significantly increased to about 51.40% under 8 μM of Smp43 pressure (Figure 5B). Moreover, pre-treatment with NAC was unable to ameliorate the apoptosis induced by Smp43, whereas CsA decreased the apoptotic cell rate caused by 4 μM of Smp43 approximately from 42.60% to 37.38% (Figure 5C). The apoptosis-induced effect of Smp43 was also confirmed via Western blot analysis. Smp43 upregulated the levels of cleaved caspase 3 with increasing concentration, accompanied by a decline in its precursors. In agreement, compared to that in control cells, Smp43 markedly upregulated the expression of Bax in the A549 cells, while the anti-apoptotic protein Bcl-1 was remarkably suppressed (Figure 5D).

### 2.6. Smp43 Interferes with Cell Cycle Distribution in A549 Cells

Due to the correlation between mitochondrial dysfunction and the cell cycle arrest of cancer cells [18], flow cytometry was conducted to confirm whether or not Smp43 exerts its antitumor capability by interrupting the cell cycle of A549 cells. After 24 h of exposure, Smp43 dose-dependently increased the quantity of A549 cells accumulated at the S phase from approximately 15.40% to 21.95%, compared with the control group (Figure 6A), while that in the G0/G1 phase was decreased from 66.50% to 60.75%. Notably, Smp43 seemed to have a slight effect on the G2/M phase of the cell cycle. In agreement with these abnormal phenomena in the cell cycle, Smp43 dose-dependently downregulated the levels of cyclin E2, cyclin A2, and CDK 2 (Figure 6B, C).

### 2.7. Smp43 Induces Autophagy in A549 Cells

Together with apoptosis, autophagy has recently become a promising strategy in cancer treatment [19]. Therefore, the autophagy-inducing activity of Smp43 in A549 cells was further investigated. Figure 7A displayed the increase in the number of autophagosomes in A549 cells after being treated with Smp43 for 12 h. Subsequently, the effect of Smp43 on the autophagy-related protein was assessed via Western blot analysis to explore its underlying mechanism in A549 cells. Under our conditions, the expressions of phosphorylated AKT and mTOR were dose-dependently suppressed in the presence of Smp43. Interestingly, the decreased levels of p62 were also observed in Smp43-treated A549 cells, indicating that autophagy was induced [20] (Figure 7B, C).

### 2.8. Smp43 Inhibits Tumor Growth in Mice

The A549 xenograft mice model was used to determine the in vivo antitumor capability of Smp43. As presented in Figure 8A, G, Smp43 treatment significantly suppressed tumor growth in terms of both volume and weight in the treated mice, compared to the control group mice. Moreover, no significant changes in body weight or important organs were observed in the Smp43-treated mice (Figure 8B–F,H). HE staining showed that the cells in the tumor tissue were damaged by Smp43, whereas the cells in the control group possessed regular shapes with visible complete structures (Figure 8I). Furthermore, immunohistochemistry staining analysis revealed the y cleaved caspase 3 induced by Smp43 (Figure 8J). These findings might suggest the potent antitumor effects of Smp43 in vivo.

## 3. Discussion

The drug resistance and cytotoxicity caused by conventional therapies in lung cancer treatment leads to the emergent need for alternatives with higher selectivity accompanied by fewer side effects. Nature-derived AMPs or designed peptides are promising alternatives due to their selectivity against cancer cells [21]. In the present study, Smp43, a venom-derived AMP from *Scorpio maurus palmatus*, was found to be able to suppress the viability of multiple lung cancer cell lines, whereas it had a slight effect on the lung fibroblast cell line MRC-5, suggesting its selective antitumor capability (Figure 1A). Interestingly, the IC_50_ value of Smp43 against A549 cells was markedly lower than that of Smp24 [7]. Furthermore, Smp43 showed antitumor capability in xenograft mice (Figure 8).

The selective antitumor activity of AMPs or anticancer peptides (ACPs) relies on the excessive anionic constituents on cancer cell membranes such as O-glycosylated mucins, phosphatidylserine, and heparan sulfates [5], whereas the outer leaflet of a normal mammalian cell primarily consists of neutral zwitterionic phospholipids. Additionally, there are some other properties of the cancer cell membrane that may enhance binding by AMPs compared to that in a normal cell, such as the more rapid phase transitions caused by the lower levels of cholesterol in the cell membrane [22] or the increasing number of microvilli in the surface area [23]. The binding reaction between AMPs and cancer cell membranes, caused by the cationic residues and the anionic constituents, leads to changes in surface charge [24]. Consistently, treatment with Smp43 caused an increase in the zeta potential of A549 cells, whereas pre-treatment with heparin markedly decreased the cellular uptake of Smp43, suggesting suppressed cellular internalization (Figure 2).

Upon binding to cancer cells, antitumor AMPs might exert their membranolytic activity through a pore-forming mechanism, resulting in cell death due to cell membrane disruption [21]. A previous study reported the pore formation capability of Smp43 in the treatment of acute leukemia cell lines [13]. In the present study, evidence from a LDH release assay and SEM analysis suggested that pore formation and membrane disruption occurred in A549 cells treated with Smp43 (Figure 3). However, research is required to confirm the pore formation in A549 cells caused by Smp43. Notably, the decreased cellular uptake of Smp43 in the presence of NH_4_Cl confirmed the responsibility of endocytosis (Figure 2C), which is similar to the antitumor activity of L-K6 against human breast cancer MCF-7 cells [25].

In cancer cell biology, the cytoskeleton contributes to different aspects, such as morphogenesis and migration [26]. The metastatic process requires the reorganization of the cytoskeleton, which involves the interaction of multiple components, including actin filaments, intermediate filaments, and microtubules [27]. Among them, while microtubules and intermediate filaments are responsible for polarized network formation and cell shape maintenance, respectively, actin filaments have been considered the main component of cell motility [28]. In the present study, treatment with Smp43 for 24 h led to a visible disorganization of F-actin (Figure 3B). The effect of the regulatory capability of the actin skeleton on mitochondrial function is well-described [29], and its disruption might cause an alteration in mitochondrial potential, leading to ROS overproduction and eventual cell death [30]. As a calcineurin inhibitor, CsA may inhibit the opening of the mitochondrial permeability transition pore via interacting with CypD, therefore ameliorating mitochondrial dysfunction [31]. Under our conditions, CsA markedly reduced the loss of mitochondrial membrane potential induced by six-hour-administered Smp43 (Figure 4A) and ameliorated apoptosis rather than NAC (Figure 5C). Notably, the secretion of ROS significantly increased after 12 h of Smp43 treatment. These findings suggest that Smp43 decreases mitochondrial potential, leading to a significant increase in the ROS content (Figure 4).

As an important organelle in cellular metabolism, the dysfunction of mitochondria might dramatically affect cell proliferation, resulting in cell cycle arrest and apoptosis [32]. Consistently, Smp43 upregulates apoptosis in A549 cells via regulating the mitochondrial apoptotic pathway (Figure 5). This effect is similar to that of the antitumor capability of Pardaxin, an AMP isolated from secretions of the Red Sea Moses sole, which targets mitochondria and induces apoptosis by activating caspase-3/7 [33]. It is well-known that the kinase activity of CDK2/cyclin E is required for gene transcription and progression into and through the S phase, while CDK2/cyclin A and CDK1/cyclin A are necessary for progression through the S phase and into the G2 phase, respectively [34]. In line with this, treatment with Smp43 significantly reduced the expression of cyclin A, cyclin E, and CDK2, leading to the accumulation of cells in the S phase of the cell cycle (Figure 6).

A growing body of evidence has proven that inducing autophagy might become a promising therapy in cancer treatment [35]. For instance, the effects of the anticancer property of curcumin on A549 cells can be reduced in the presence of autophagy inhibitors [36]. LL-37-derived peptide Fk-16 could inhibit the proliferation of colon cancer cells by inducing apoptosis and autophagy through the p53-BCl-2/Bax cascade reaction [37]. It has been reported that the p62 protein is continuously degraded in the cytoplasm during autophagy, which is consistent with our results from the Western blot analysis (Figure 5D). Moreover, the role of Bcl-2 in anti-autophagy was well described [38]. Bcl-2 can bind to Beclin 1, a crucial determining factor in both autophagy and apoptosis processes [39], preventing the assembly of the pre-autophagosomal structure, which results in autophagy inhibition. Thus, it is reasonable to believe that the decrease in Bcl-2 levels caused by Smp43 might co-exist with the increase in Beclin 1, which is consistent with the antitumor mechanism of licorice and licochalcone-A against human LNCaP prostate cancer cells [40]. However, research is required to confirm the change in Beclin 1 expression in A549 cells caused by Smp43. The inhibition of Akt/mTOR might involve both apoptotic and autophagic cell death as well as the suppression of cell migration [41,42,43]. In the present study, treatment with Smp43 resulted in the repression of the activation of the Akt/mTOR signaling pathways, which is in line with the antitumor capability of gefitinib against A549 cells via exerted via the inducement of apoptosis and autophagy [44].

The cellular selectivity of Smp43 toward cancer cells and its low toxicity in vivo demonstrate its potential as an antitumor agent. However, peptide drugs have faced multiple challenges in clinical trials, including non-specific cytotoxicity, in vivo conformational instability, immunogenicity, and high production costs [45]. Therefore, designing novel peptide derivatives with fewer residues and improved activity has become an interesting research field. Smp43(1–14) is a unique fragment among a series of 14-residue fragments of Smp43 that exerts enhanced antimicrobial activity compared to its parent peptide by integrating into the plasma membrane of bacteria and inducing the formation of pores [46]. Further, CD spectra analysis and Psipred prediction suggest that Smp43(1–14) adopts a helical structure [47]. Considering that Smp43 inhibits tumors via membrane lysis, it is reasonable to assume that Smp43 (1–14) is also active against cancer cells and may have more advantages due to its shorter sequence. However, further research is necessary to confirm this hypothesis.

## 4. Conclusions

In conclusion, the present study elucidates the antitumor mechanism of Smp43 against A549 cells. Smp43 enters A549 cells through endocytosis, leading to the collapse of the cell membrane, disorganization of the cytoskeleton, and mitochondrial dysfunction. This, in turn, leads to abnormal cell cycle distribution and various forms of cell death, including necrosis, apoptosis, and autophagy. The peptide’s in vivo antitumor efficacy was demonstrated using a xenograft assay. These findings highlight the potential of Smp43 as a therapy for non-small-cell lung cancer.

## 5. Materials and Methods

### 5.1. Reagents and Cell Culture

Cell culture mediums (RPMI-1640 and Dulbecco’s modified Eagle’s medium), trypsin, fetal bovine serum (FBS), and phosphate-buffered saline (PBS) were purchased from Gibco (Grand Island, NY, USA). Lung cancer cell lines (H460, H3122, PC9, and A549) and human fetal lung fibroblast cells, MRC-5 cells, were purchased from ATCC (Manassas, VA, USA). Cells were cultured in the corresponding medium containing 1% of the dual antibiotic solution penicillin–streptomycin with 10% FBS (37 °C, 5% CO_2_). Bax, Bcl-2, caspase-3, cleaved caspase-3, CDK2, cyclin A2, cyclin E2, mTOR, Akt, p-Akt, p-mTOR, p62, GAPDH, and the corresponding secondary antibodies were purchased from Cell Signaling Technology (Beverly, MA, USA). DAPI, N-Acetyl-L-cysteine (NAC), and different commercial kits used in multiple assays including reactive oxygen species (ROS) production, EdU, JC-1 staining, apoptosis, and cell cycle analysis were purchased from Beyotime Institute of Biotechnology, China. Smp43 and FITC-labeled Smp43 were synthesized as mentioned in our reported research [13].

### 5.2. Animals and Ethics Statement

BALB/c nude mice at 6–8 weeks old with approximately 20 g of body weight were obtained from the Laboratory Animal Center of Southern Medical University. The mice were randomly and separately housed in groups of five under specific pathogen-free conditions with a 21 ± 2 °C room temperature, 60% humidity, and a 12 h light–dark cycle. The study protocol was approved by the Animal Ethics Committee of Southern Medical University (no. L2019226).

### 5.3. Cytotoxicity Assays

The cytotoxicity of Smp43 against multiple cell lines was evaluated using a MTT assay as reported in our previous study [14]. In brief, after being placed in 96-well plates overnight, cells with a density of 1 × 10^4^ cells per well were treated with Smp43 at different concentrations (2.5, 5, 10, and 20 μM) for 12 h and 24 h. Cells were subsequently incubated with 10 µL of MTT (5 mg/mL) in the dark for 4 h at 37°C. The cell medium was replaced by 200 of µL DMSO, and the optical density at 570 nm was determined with a microplate reader (Tecan Company, Männedorf, Switzerland).

The EdU assay was performed using a commercial kit to assess the effect of Smp43 on A549 cell proliferation. Briefly, A549 cells (2 × 10^5^ cells/well) were treated with a sequence of concentrations of Smp43 (2–8 μM) for 24 h in 6-well plates. Samples were subsequently treated with EdU and Alexa Fluor 488 for 2 h and 30 min, respectively, in the dark. The changes in fluorescence were observed using fluorescence microscopy. All experiments were accomplished in triplicate.

### 5.4. Cell Morphology Evaluation

A549 cells (2 × 10^5^ cells/well) were cultured overnight in a six-well plate and subsequently incubated with Smp43 at different concentrations (2–8 μM) for 24 h. An inverted phase-contrast microscope (CKX41, Olympus, Tokyo, Japan) was used to observe the changes in cell morphology. Three to five photographs of each well were taken.

### 5.5. Colony Formation Assessment

A549 cells at a density of 500 cells in each well were cultured overnight in a six-well plate. Smp43 at different concentrations (2–8 μM) was administered for 8 days. On the last day of the experiment, the cell medium was discarded, and the cells were fixed with methanol for 10 min at room temperature. Crystal violet dye (Beyotime, Shanghai, China) was used for colony staining and samples were observed using an ELISPOT analyzer (S6 Versa, CTL, NY, USA).

### 5.6. Peptide Internalization Assessment

To evaluate the location of Smp43 in A549 cells after different times of treatment (6, 12, and 24 h), the cells were treated with 4 μM FITC-labeled Smp43 at 37 °C and subsequently washed with PBS. A concentration of 4% of paraformaldehyde (PFA) was used to fix the washed cells for 30 min, followed by 10 min of DAPI staining. Samples were eventually observed using a fluorescence microscope. Three to five photographs of each well were taken.

The changes in the internalization of Smp43 into A549 cells in the presence of heparin were evaluated using flow cytometry analysis. In brief, 4 µM of FITC-labeled Smp43 was co-incubated with 5 µg/mL of heparin for 30 min in a RPMI-1640 medium. The mixture was then co-incubated with A549 cells at a density of 1 × 10^5^ cells each well in a six-well plate for 6 h, and flow cytometry was performed to measure cell fluorescence.

The role of the cellular energy state in the internalization of Smp43 was also assessed. After being pre-incubated at different temperatures (37 °C and 4 °C) for 30 min, A549 cells were subsequently treated with 4 µM of FITC-labeled Smp43 for 6 h, and the changes in the cellular uptake of Smp43 were identified via flow cytometry. In another set of experiments, A549 cells were incubated with 50 mM NH_4_Cl for 30 min, and subsequently treated with 4 µM of FITC-labeled Smp43 for another 6 h before flow cytometry analysis. All experiments were conducted in triplicate.

### 5.7. Zeta Potential Assessment

The binding reaction between the peptide and cancer cell membrane was evaluated via zeta potential calculation. Briefly, A549 cells at a density of 1 × 10^5^ were treated with different concentrations of Smp43 (2–16 µM) at room temperature for 10 min. The sample’s zeta (ξ) potential was measured using the Folded Capillary cell (DTS1070) and the Zetasizer system (Nano ZS; Malvern Instruments Ltd., Worcestershire, UK). All experiments were performed in triplicate.

### 5.8. LDH Release Assay

The changes in the LDH release rate from A549 cells caused by Smp43 were measured using a commercial kit in accordance with the manufacturer’s instructions (Beyotime, Shanghai, China). Briefly, a sequence of concentrations of Smp43 (2–16 μM) was used to treat A549 cells for 12 h and 24 h, respectively. Samples were then incubated with 10 µL of the LDH release solution for 1 h. The supernatants were collected and incubated with 60 µL of a substrate solution for 30 min in the dark. The mixture’s optical density at 490 nm was obtained using a microplate reader (Tecan Company, Männedorf, Switzerland). All experiments were performed in triplicate.

### 5.9. Scanning Electron Microscope Examination

To observe the disruption of the A549 cell membrane caused by Smp43, A549 cells at a density of 2 × 10^5^ cells in each well were cultured overnight in a six-well plate. After being treated with Smp43 (4 and 8 μM) for 12 h, samples were fixed overnight with 2.5% glutaraldehyde in a 0.1 M buffer, washed three times with PBS, and dehydrated with a series of gradient ethanol/water solutions. After drying and gold-coating, the changes in the cell membrane were observed with the Phenom ProX instrument at 15 kV and an 8000× magnification. About 5 to 10 photographs were randomly taken.

### 5.10. Fluorescence Microscopy Determination

The changes in cytoskeleton organization, mitochondrial potential, ROS secretion, and DAPI staining were observed using fluorescence microscopy. A549 cells at a density of 1 × 10^5^ cells in each well were cultured overnight in 12-well plates before being treated with Smp43 (4 and 8 μM) for 24 h. The PBS-washed cells were fixed with 4% PFA for 10 min. After being rinsed in PBS at least three times, and 5 min of 0.2% Triton X-100 permeabilization, samples were labeled with rhodamine–phalloidin (1:100, Sigma-Aldrich; Darmstadt, Germany) at room temperature for 30 min. Labeled cells were incubated with DAPI (Beyotime, Shanghai, China) for 10 min with protection from light at room temperature, washed three times with PBS, and the changes in F-actin were observed via fluorescence microscopy (Axio Observer, Zeiss, Oberkochen, Germany).

The levels of cellular ROS were assessed using 2′,7′-dichloro-dihydro-fluorescein diacetate (DCFH-DA) staining. A549 cells (5 × 10^4^ cells/well) were cultured overnight in a 24-well plate overnight before being treated with Smp43 (2, 4, and 8 μM) for 6 and 12 h, followed by 10 μM of DCFH-DA staining for 30 min with protection from light at 37°C. Samples were washed three times with PBS before being photographed.

The changes in mitochondrial membrane potential were determined with a commercial JC-1 staining kit. Briefly, A549 cells were treated with Smp43 at different concentrations (2–8 μM) for 6 h. The collected cells were washed with PBS, re-suspended in the JC-1 working solution (500 μL), and incubated for 20 min at 37 °C with 5% CO_2_. Subsequently, the cells were harvested, washed, and re-suspended in a 1× incubation buffer two times. The changes in mitochondrial potential caused by Smp43 were observed. Cyclosporin A (CsA) was used as the positive control.

A DAPI staining assay was performed to evaluate the apoptosis induced by Smp43 in A549 cells. After being permeabilized with 0.1% Triton X-100 at 4 °C for 2 min, Smp43-treated cells were incubated with DAPI and observed using fluorescence microscopy. About three photographs of each group in experiments were randomly obtained.

### 5.11. Flow Cytometry

Flow cytometry was performed to assess the overproduction of ROS in A549 cells caused by Smp43, and to determine whether or not Smp43 affects the apoptosis process and cell cycle distribution of A549 cells. A549 cells (2 × 10^5^ cells/well) were cultured overnight in six-well plates and treated with different concentrations of Smp43 (2, 4, and 8 μM), 2 mM of NAC or 2 μM of CsA for 24 h, respectively. For apoptosis assessment, the PBS-washed A549 cells were stained with annexin V-FITC and PI for 15 min without light before flow cytometry analysis. In another set of experiments, collected cells were fixed with 70% ethanol overnight at 4 °C followed by propidium iodide (PI) incubation at room temperature for 30 min to assess the effect of Smp43 on cell cycle distribution.

For the ROS overproduction analysis, A549 cells were stained with DCFH-DA as mentioned above, and were washed three times with PBS before being analyzed. FlowJo (ver.8.4) was used for data analysis, and all experiments were conducted in triplicate.

### 5.12. Transmission Electron Microscopy Determination

A549 cells (2 × 10^5^ cells/well) were grown on a six-well plate for 24 h, followed by 24 h of 4 μM Smp43 co-incubation. Cells without Smp43 treatment were considered the negative control. The harvested A549 cells were re-suspended in the TEM fixative, and then fixed at 4 °C for 2 h. Subsequently, the cells were collected via centrifugation, the supernatants were discarded, and the precipitations were washed with 0.1 M PB (pH = 7.4) three times for 3 min. A concentration of 1% of a prepared agarose solution was added for pre-embedding. After being fixed with 1% OsO_4_ in 0.1 M PB (pH 7.4) for 2 h at room temperature, samples were rinsed in 0.1 M PB (pH 7.4) 3 times, 15 min each, followed by being dehydrated at room temperature with a gradient of ethanol. Samples were inserted into the pure EMBed 812 (SPI, West Chester, IL, USA), and the removed resin blocks from the embedding models were cut until they were 60–80 nm thin on the ultra-microtome (Leica Biosystems, Nussloch, Germany) and fished out onto the 150-mesh cuprum grids with formvar film. After being stained with a 2% uranium acetate-saturated alcohol solution and 2.6% lead citrate, the cuprum grids were observed under a TEM (Hitachi, Tokyo, Japan).

### 5.13. Animal Studies

A549 cells (5 × 10^6^ cells) with 90% of viability were subcutaneously administrated into the right flank of each experimental individual. A caliper was used daily to record and evaluate the changes in tumor size together with the palpation method. The following formula was used to define the tumor volumes: volume (mm^3^) = (smallest diameter)^2^ × (largest diameter)/2. After 18 days of injection, either 2 mg/kg of Smp43 or physiological saline was injected every three days near the tumor site. After 18 days of treatment, the tumor tissues and multiple organs were surgically removed and weighed. The morphological changes in lung cancer cells caused by Smp43 were evaluated via H&E staining after being embedded in paraffin. The apoptosis levels of tumor tissue were further detected via immune-histochemical staining, and the alterations in the apoptosis-related cleaved caspase-3 were observed using an inverted phase-contrast microscope. Data were acquired for five individual mice in each group.

### 5.14. Western Blot Analysis

A549 cells (2 × 10^5^ cells/well) were cultured overnight in a six-well plate and subsequently treated with Smp43 (2–8 μM) for 24 h. The harvested cells were washed with ice-cold PBS, and the proteins were extracted as mentioned in our reported study [11]. Primary antibodies against Bax, Bcl-2, caspase-3, cleaved caspase-3, cyclin A2, cyclin E2, CDK2, p62, Akt, p-Akt, p-mTOR, mTOR, and GAPDH (4 °C, overnight) and the corresponding secondary antibodies (room temperature, 1 h) were used in the Western blot analysis. The blot bands were observed using a FluorChem R system (ProteinSimple, San Jose, CA, USA). ImageJ was used for band density quantification and all experiments were conducted in triplicate.

### 5.15. Statistical Analysis

GraphPad Prism 8.0 (GraphPad Software, Inc., La Jolla, CA, USA) was used to perform statistical analysis, in which a one-way ANOVA with Bonferroni’s multiple comparisons was applied. Data were presented as mean ± SD and * *p* < 0.05, ** *p* < 0.01, and *** *p* < 0.001 were considered statistically significant.

## Figures and Tables

**Figure 1 toxins-15-00347-f001:**
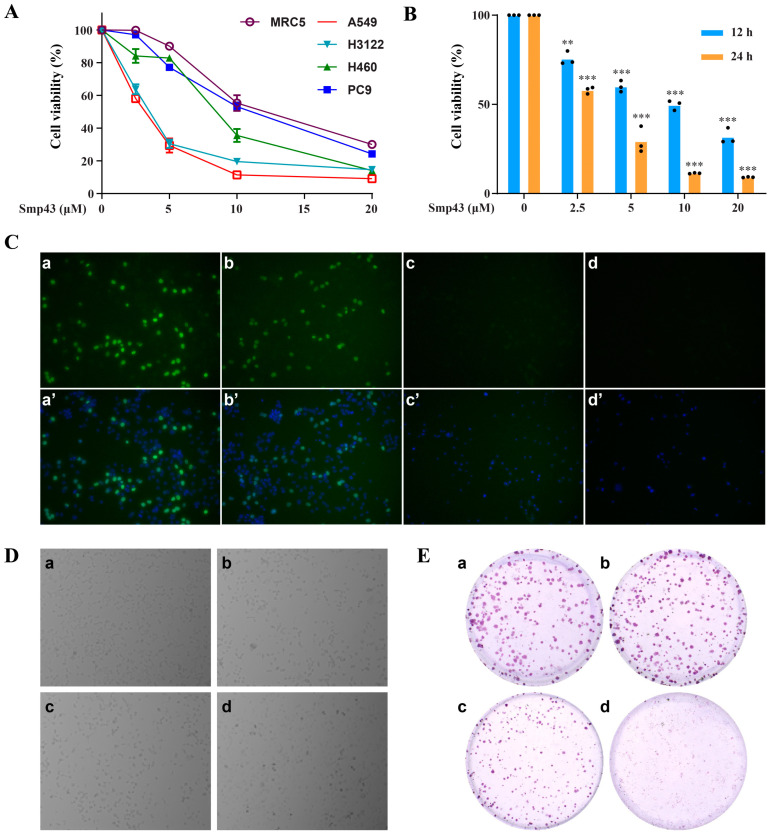
Cytotoxicity of Smp43 on A549 cells. (**A**) Effect of Smp43 (0–20 μM) against different human lung cancer cell lines and MRC-5 cells. MTT assay was conducted after 24 h of treatment. (**B**) Cytotoxic effect of different concentrations of Smp43 on A549 cells. MTT assay was performed after 12 h and 24 h of Smp43 treatment. The dots represent the individual data points. (**C**) EdU staining assay. Panels (**a**–**d**) express EdU-stained A549 cells, and EdU-DAPI staining merged images are presented in panels (**a’**–**d’**). The control cells are presented in panels (**a**,**a’**); panels (**b**–**d**,**b’**–**d’**) display A549 cells incubated with 2, 4, and 8 μM of Smp43, respectively. Cells were photographed using 200× magnification fluorescence microscopy. (**D**) Cell morphology changes caused by Smp43. The control cells are presented in panel (**a**); panels (**b**–**d**) display A549 cells incubated with 2, 4, and 8 μM of Smp43, respectively. Photographs were obtained at 40× magnification. (**E**) Colony formation assay. The control cells are presented in panels (**a**); panels (**b**–**d**) display A549 cells in the presence of 2, 4, and 8 μM of Smp43, respectively. Data are represented as the mean ± SD (*n* = 3). The significant differences between the treated and control group are presented as ** *p* < 0.01 and *** *p* < 0.001.

**Figure 2 toxins-15-00347-f002:**
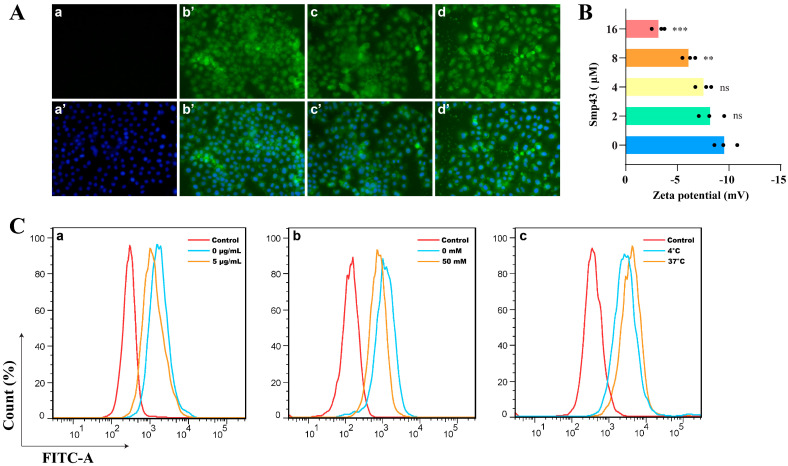
Smp43 is internalized into A549 cells. (**A**) Detection of the amount of Smp43 entering A549 cells. The upper panel represents A549 cells stained with 4 μM of FITC-labeled Smp43, and the merged images of cells stained with FITC-labeled Smp43 and DAPI are displayed in the lower panel. The control cells are presented in panels (**a**,**a’**); the cells treated with Smp43 at different time points (6, 12, and 24 h) are shown in panels (**b**–**d**,**b’**–**d’**), respectively. Photographs were obtained at 400× magnification. (**B**) Zeta potential assay. The variation in membrane potential was observed after the cells were co-incubated with Smp43 for 10 min, and the control group was treated with PBS. Data from A549 cells treated with different concentrations of Smp43 were presented in different colors. (**C**) Effects of heparin (panel (**a**)), NH_4_Cl (panel (**b**)), and different temperatures (panel (**c**)) on the amount of Smp43 entering A549 cells. The action time was 6 h, and cells without any treatment were considered control. Data are represented as the mean ± SD (*n* = 3). The significant differences between the treated and control group are presented as ** *p* < 0.01 and *** *p* < 0.001. “ns” stands for not significant (*p* > 0.05).

**Figure 3 toxins-15-00347-f003:**
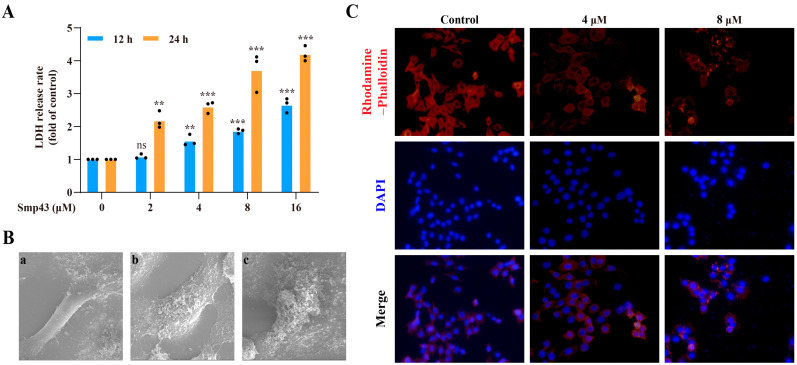
Smp43 induces cell membrane disruption and alters cytoskeleton organization. (**A**) LDH release rate from A549 cells in the presence of Smp43. Data are represented as the mean ± SD (*n* = 3). (**B**) SEM analysis. A549 cells were observed after being treated with Smp43 for 12 h, and PBS was used in the control group. Panel (**a**): the control cell; panels (**b**,**c**): the cell treated with 4 and 8 μM of Smp43, respectively. (**C**) The changes in cytoskeleton organization caused by Smp43. Photographs were obtained using fluorescence microscopy at 400× magnification. The significant differences between the treated and control group are presented as ** *p* < 0.01and *** *p* < 0.001. “ns” stands for not significant (*p* > 0.05).

**Figure 4 toxins-15-00347-f004:**
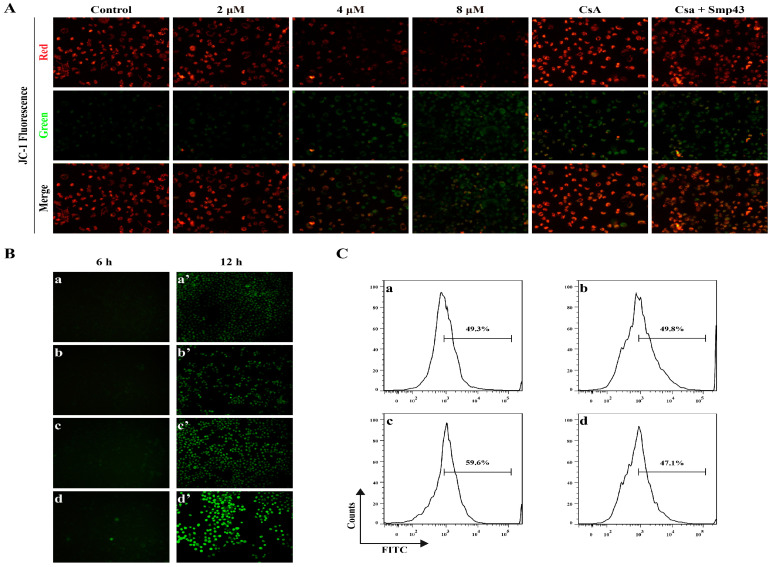
Smp43 damages mitochondria and induces ROS accumulation. (**A**) The changes in mitochondrial membrane potential in the presence of Smp43. A549 cells were incubated with Smp43 (2–8 μM) for 6 h. Positive control CsA was co-incubated with A549 cells for 1 h before administration of 4 μM of Smp43. Images were obtained at 200× magnification. (**B**) The overproduction of ROS in A549 cells after 6 and 12 h of Smp43 treatment is displayed in the left and right panels, respectively. The control cells are presented in panels (**a**,**a’**); the cells treated with Smp43 (2, 4, and 8 μM) are shown in panels (**b**–**d**,**b’**–**d’**), respectively. Photographs were obtained at 100× magnification. (**C**) The effect of NAC on the intracellular ROS generation induced by Smp43. Panel (**a**): the control cells without any treatment; the cells treated with NAC or Smp43 are presented in panels (**b**,**c**), respectively; panel (**d**): A549 cells were treated with both NAC and Smp43. DCFH-DA stained A549 cells were pre-treated with 2 mM of NAC for 1 h before being treated with Smp43 for 12 h.

**Figure 5 toxins-15-00347-f005:**
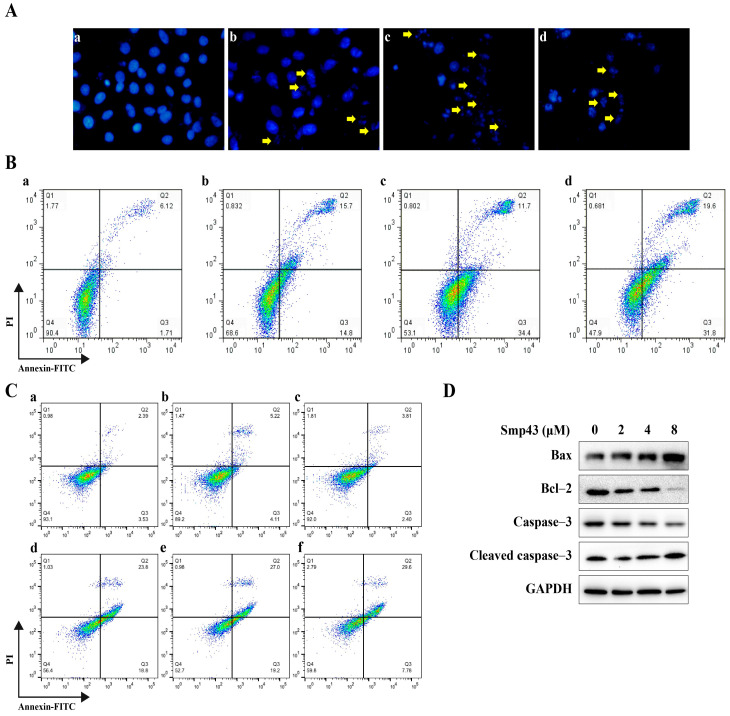
Smp43 induces apoptosis in A549 cells. (**A**) DAPI staining assay. Different concentrations of Smp43 were co-incubated with A549 cells for 24 h before DAPI staining; the control group was treated with PBS only. The control cells are presented in panel (**a**); Smp43-treated cells (2, 4, and 8 μM) are shown in panels (**b**–**d**), respectively. Photographs were obtained at 400× magnification. Yellow arrows represent apoptotic bodies. (**B**) Flow cytometry analysis. A549 cells were pre-treated with Smp43 for 24 h before being stained by annexin V-FITC/PI. The control cells are presented in panel (**a**); Smp43-treated cells (2, 4, and 8 μM) are shown in panels (**b**–**d**), respectively. (**C**) The changes in Smp43-induced apoptosis in the presence of NAC and CsA. A549 cells were incubated with NAC (2 mM) or CsA (2 μM) for 1 h before treatment with 4 μM of Smp43; the control group was treated with PBS only. The control cells are presented in panel (**a**); the cells treated with NAC or CsA only are shown in panels (**b**,**c**), respectively; A549 cells treated with Smp43 only, NAC + Smp43, and CsA + Smp43 are displayed in panels (**d**–**f**), respectively. (**D**) Western blot analysis of the apoptotic pathway in A549 cells.

**Figure 6 toxins-15-00347-f006:**
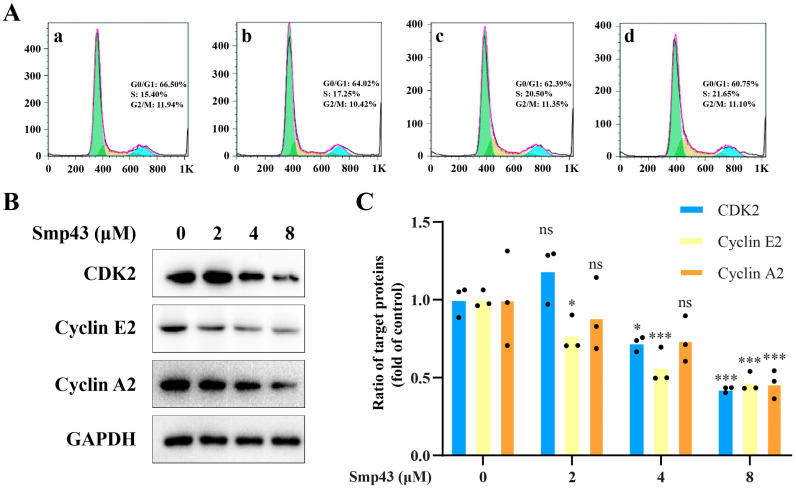
Smp43 interferes with cell cycle distribution in A549 cells. (**A**) Cell cycle distribution analysis via flow cytometry. The control cells are presented in panel (**a**); the cells treated with Smp43 (2, 4, and 8 μM) are shown in panels (**b**–**d**), respectively. (**B**) Western blot analysis of Smp43-treated A549 cells. (**C**) Quantification of the Western blot assay. Data are represented as the mean ± SD (*n* = 3). The significant differences between the treated and control group are presented as * *p* < 0.05 and *** *p* < 0.001. “ns” stands for not significant (*p* > 0.05).

**Figure 7 toxins-15-00347-f007:**
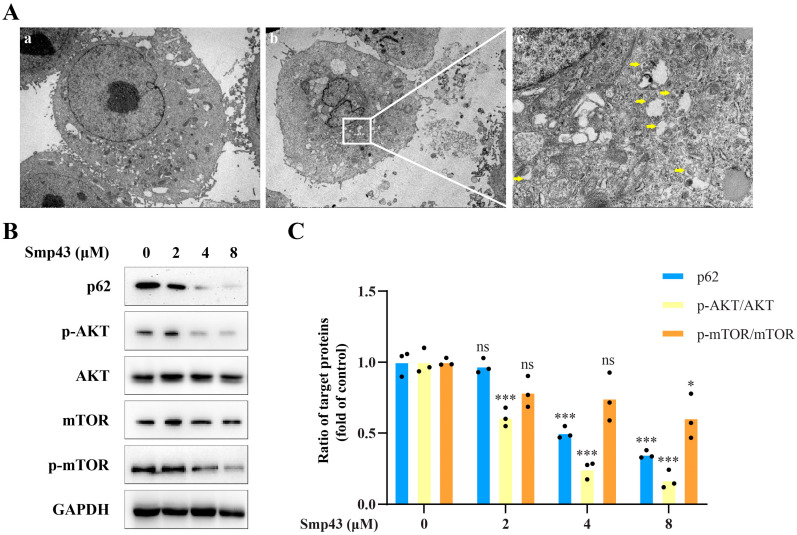
Smp43 induces autophagy in A549 cells. (**A**) The changes in A549 cell morphology caused by Smp43. The control cells and Smp43-treated A549 cells are presented in panels (**a**,**b**), respectively; panel (**c**): the enlarged image of the corresponding area in panel (**b**). Images were obtained via an electron microscope and the autophagosomes are highlighted by yellow arrows. (**B**) The expression of autophagy-related protein in Smp43-treated A549 cells. (**C**) Quantification of the Western blot assay. Data are represented as the mean ± SD (*n* = 3). The significant differences between the treated and control group are presented as * *p* < 0.05 and *** *p* < 0.001. “ns” stands for not significant (*p* > 0.05).

**Figure 8 toxins-15-00347-f008:**
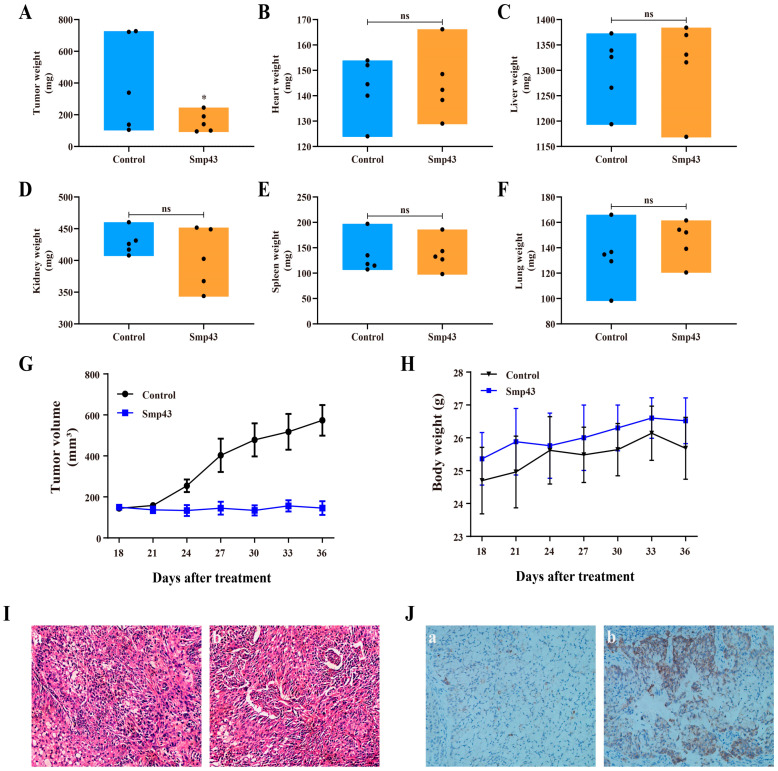
The in vivo antitumor effect of Smp43. (**A**) Tumor weight after 18 days of Smp43 treatment. (**B**–**F**) The changes caused by Smp43 on the heart, liver, kidney, spleen, and lung, respectively. (**G**) Tumor volume changes caused by Smp43. (**H**) The changes in body weight of mice. (**I**) Pathological images of tumor tissue obtained via H&E staining from control (**a**) and Smp43-treated mice (**b**). (**J**) Cleaved caspase 3 immunohistochemistry in control (**a**) and Smp43-treated mice (**b**). Photographs were obtained at 200× magnification. Data are represented as the mean ± SD (*n* = 3). The significant difference between the treated and control group is presented as * *p* < 0.05. “ns” stands for not significant (*p* > 0.05).

## Data Availability

All data are available on request.

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
