# Peer review of "The Potent Antitumor Activity of Smp43 against Non-Small-Cell Lung Cancer A549 Cells via Inducing Membranolysis and Mitochondrial Dysfunction"

_toxins, 2023, doi:10.3390/toxins15050347_

Round 1

Reviewer 1 Report

In this article the Authors study the anticancer activity of a Smp43 peptide derived from a scorpion’s venom. The study is mostly designed and performed in a decent manner. Regarding the figures, I would rather use boxes with whiskers or scatter dot plots instead of bar charts. The bar charts are commonly used for most scientific reports, but they are statistically inappropriate since they do not show the variability of data points within the sample. From the statistical point of view, the bar charts suggest that the values of the data points within the sample vary from 0 (bottom of the bar) to the value shown by the top of the bar (which is actually the arithmetic mean within the group). I have also found some other major and minor issues listed below.

Major issues:

- the Authors should use a non-cancerous cell line for all in vitro experiments, not only for MTT assay – in my opinion, all the experiments that the Authors did with A549 cells should be performed additionally with an appropriate non-cancerous control cell line (see below for details);

- the Authors used MRC-5 lung fibroblasts as non-cancerous control cell line; although MRC-5 fibroblasts are of lung origin, there are cell lines more suitable as a “normal” control for A549 cells, including: Primary Human Small Airway Epithelial Cells (HSAEC) or Human Primary Bronchial/Tracheal Epithelial Cells;

- SEM (Standard Error of The Mean) can be used only if the number of replicates is equal or greater than 100; the Authors, however, used the minimum number of replicates required for statistical analysis – 3 – contributing to the low statistical power of the estimated interferences; for sample sizes smaller than 100 replicates the SD (Standard Deviation) should be used instead – the Authors should fix this basic mistake commonly found in statistical analyses performed for numerous scientific papers;

- it would be better to repeat each experiment more than 3 times, to increase the number of replicates and to provide the greater statistical power of experimental data.

Minor issues:

- the list of abbreviations is necessary: the Authors use a great number of abbreviations and it is hard to remember each of them during reading, some abbreviations are not explained (for example ‘ns’ over the bars in Figure 6C – probably ‘not significant’);

- how many times the Zeta potential assessments were performed? The triplicate was applied for the same experiment, or the whole experiment was repeated three times?

- the description for Figure 3B says: “Photographs were obtained using fluorescence microscopy”; looking at the images I would rather say, that a confocal microscope was used – am I right?

- Figure 4A – the columns are described, whereas the rows – not; there are the images organized in 3 rows and it is not written anywhere, what is presented by these rows;  

- Figure 4C – the panels a, b c and d are not described anywhere – it makes this figure to be impossible to understand;

- Figure 6C is not described anywhere in a text body of the manuscript, it is shown only in the description to the figure – the comments in the main text are missing, and the abbreviation ‘ns’ is not explained (I can only guess, that it means “not significant”, but everything must me clearly described in a scientific paper);

- Figure 7A – it is worth to mention in the description to this figure, that the images were obtained by an electron microscope;

- Figure 8B – Smp43 did not exert any changes on tumor volume, thus the description should be changed, for example “The effect of Smp43 on tumor volume”;

- page 4, line 96: change “concentration-dependently” to “in a concentration dependent manner” – it would be more elegant;

- page 4, line 97-98: change “the interaction of Smp43 to the 97 cancer cell membrane” to “the interaction of Smp43 with the 97 cancer cell membrane”;

- page 4, line 105: change “heparin sulfate” to “heparin sulfate”;

- page 11, line 266: low level of cholesterol enhances the fluidity of the membrane only over the transition temperature, when the ambient temperature is lower than transition temperature, the membrane fluidity is decreased due to low cholesterol levels; simply saying, the presence of cholesterol protects the cell membrane from the changes in fluidity due to temperature changes, thus low cholesterol levels cause the faster changes in membrane fluidity;

- page 15, line 508: there is an extra “lonely” letter “t” to be removed.

There are only minor language mistakes and all of them are listed in the Comments and Suggestions for Authors:

- page 4, line 96: change “concentration-dependently” to “in a concentration dependent manner” – it would be more elegant;

- page 4, line 97-98: change “the interaction of Smp43 to the 97 cancer cell membrane” to “the interaction of Smp43 with the 97 cancer cell membrane”;

- page 4, line 105: change “heparin sulfate” to “heparin sulfate”.

Author Response

Dear Reviewer,

We deeply thank you for your precious time in reviewing our paper and providing valuable comments. We have carefully considered the comments and tried our best to address every one of them. Here, we provide a point-to-point response.

Thank you for your patience. 

Reviewer 2 Report

The current work is devoted to scorpion venom derived peptide Smp43 (43 is the length of the peptide): GVWDWIKKTAGKIWNSEPVKALKSQALNAAKNFVAEKIGATPS. Specifically, anticancer activity of the peptide was evaluated against NSCLC. This work is a continuation of the previous study (ref. 12: Front. Pharmacol. 2021, 12, 788874), where anticancer effect of the peptide towards two acute leukaemia cell lines (myeloid (KG1-a) and lymphoid (CCRF-CEM) leukaemia cell lines) has been revealed. The present work describes numerous in vitro and in vivo experiments, thus extensively characterising anticancer effect of the peptide against NSCLC.

However, it is not clear why the peptide was used as is, without any modification. Another peptide, Smp24 also exhibited anticancer properties (ref. 10).

Major concern:

Why Smp43 was not shortened? What is the role of acidic residues: Asp4 and Glu17? In the paper it has been noted that the plasma membrane of cancer cells is acidic. Thus, the residues with negative charge are not favourable for the activity. Nothing is known about spatial structure of Smp43 peptide. It is difficult to imagine that the peptide is a continuous helix 1-43. Probably, it constitutes a combination of several helical regions. Moreover, I found a work where a shortened fragment 1-14 of Smp43 was used: https://pubmed.ncbi.nlm.nih.gov/34064808/. May be, this peptide is also active against cancer cells? I think that adding a text discussing possible modifications of Smp43 which might increase its anticancer properties is warranted in the Discussion section. Alternatively, refer to the works where this question has been discussed.

Minor notions (corrections in the text of the manuscript are required):

Line 25: 75-80% of all cases...You mean: of all lung cancer cases?

Line 34: a high proportion of hydrophobic residues… In AMP this proportion varies from ~30 to ~80%. Thus the statement is disorienting. Please, either say exact estimates, or indicate the range.

Lines 42-44: References should be provided in the end of this long sentence.

Line 50: cancer… Probably, you meant “anticancer”.

Line 125,138,141,439 : cytoskeleton conformation...Better – organization.

Line 171: picture 5A → Figure 5a.

Line 176: significantly decreased ...from 42.60% to 37.38%; This is not significant change, please, remove the word “significantly”.

Lines 178-179: Smp24 upregulated the levels of cleaved-caspase 3 with raising concentration, accompanied by its precursors declined. - If you refer to another peptide, Smp24, please add the respective reference.

Line 202: significantly decreased from 66.50% to 60.75%...This is not significant variation. Please, remove the word “significantly”.

Line 341: Smp43 internalizes into A549 cells in a pore formation manner… However, in the preceding text (lines 278-279) it has been stated: “However, research is required to confirm the pore formation in A549 cells caused by Smp43.” Please, remove the line 341 from Conclusions, or make this statement softer, introducing words “most likely”.

Lines 387, 400: About three photographs of each well were taken. Please, say definitely, how many photographs, e.g. Not less than three, or At least, three…

Too many errors. Extensive editing is required.

Author Response

Dear Reviewer,
We deeply thank you for your precious time in reviewing our paper and providing valuable comments. We have carefully considered the comments and tried our best to address every one of them. Here, we provide a point-by-point response.
Thank you for your patience. 

Round 2

Reviewer 1 Report

Dear Authors,

with the great satisfaction I accept all of your corrections. In the present form of the manuscript the graphs are correctly designed and SEM was changed to SD as suggested. I can also accept your explanation regarding the MRC-5 cells to be used as non-tumorigenic ("normal") control.

I would like to explain just two minor issues that still remain to be corrected:

- due to my spelling mistake one of my suggestions was not understood; I should have written: page 4, line 105: change “heparin sulfate” to “heparan sulfate”;

- regarding the fragment about cholesterol on page 11, line 266, I would change it as follows: "the more rapid phase transitions caused by the lower levels of cholesterol in the cell membrane" - because the low level of cholesterol increases the membrane fluidity only above the transition temperature - the dependence between the membrane fluidity and the cholesterol content is not as simple as the Authors of the reference no. 21 suggest.

In a summary, I will accept your manuscript with all the revisions that you already made plus these two minor corrections mentioned above. 

With warm regards to the Authors.

Author Response

Dear Reviewer,

Once again, we appreciate your concern about our manuscript and all the constructive comments. We re-checked again and realized some mistakes in our manuscript. The reagent used in the present study is "heparin" instead of "heparin sulfate". Due to the structural relationship between "heparin" and "heparan sulfate", "heparin" was used as a competitor. All the "heparin sulfate" words were replaced by "heparin", and the word "heparin sulfate" used in the "Discussion" section was substituted by "heparan sulfate" as you recommended.

The fragment about cholesterol on page 11 was also re-wrote as you suggested. 

All the changes were highlighted in the new version of the revised manuscript. Please do not hesitate to contact us if you have any questions.

Thank you for your precious time in supporting us.

Kind regards.

Reviewer 2 Report

If you will test anticancer properties of Smp(1-14), I recommend to check also longer peptides, e.g (1-18), (1-22)...etc. So, the length effect will be seen clearly.

Author Response

Dear Reviewer,

Once again, we appreciate your concern about our manuscript and all the constructive comments about the modification strategy of Smp43. 

Kind regards.